# Nonlinear vortex solution for perturbations in the Earth's Ionosphere

Miroslava Vukcevic[1] and Luka Č. Popović[1]

[1]Astronomical Observatory Belgrade, Volgina 7, Belgrade, Serbia

**Correspondence:** M. Vukcevic(vuk.mira@gmail.com)

**Abstract.** There are many observational evidences of different fine structures in the ionosphere and magnetosphere of the Earth. Such structures are created and evolve as a perturbation of the ionosphere's parameters. Instead of dealing with number of linear waves, we propose to investigate and follow up the perturbations in the ionosphere by dynamics of soliton structure. Apart of the fact that it is more accurate solution, the advantage of soliton solution is its localization in space and time as consequence of balance between nonlinearity and dispersion. The existence of such structure is driven by the properties of the medium. We derive necessary condition for having nonlinear soliton wave, taking the vortex shape, as description of ionosphere parameters perturbation. We employ magnetohydrodynamical description for the ionosphere in plane geometry, including rotational effects, magnetic field effects via ponderomotive force, pressure and gravitational potential effects, treating the problem self-consistently and nonlinearly. In addition, we consider compressible perturbation. As a result, we have obtained that Coriolis force and magnetic force at one side, and pressure and gravity on the other side, determine dispersive properties. Dispersion at higher latitudes is mainly driven by rotation, while near the equator, within the E and F-layer of ionosphere, magnetic field modifies the soliton solution. Also, very general description of the ionosphere results in the conclusion that the unperturbed thickness of the ionosphere layer cannot be taken as ad hoc assumption, it is rather consequence of equilibrium property, which is shown in this calculation.

## 1 Introduction

The structure of the Earth's ionosphere depends on its distance from the Earth surface involving different sources of disturbance in basic parameters within it. These sources could be subject of events in Earth, atmosphere, Sun and $\gamma$-ray bursts (GRB) from deep space. Observation evidence of large scale ionosphere structures is important as interpretation of the low frequency perturbations of the ionosphere response to mentioned disturbance sources. That interpretation is an important research task, since the studies of perturbations induced in ionosphere can be used in different fields related to human life, for example, prediction of natural disasters (atmospheric storms, volcano eruptions, earthquakes) or problems in satellites and electrical devices operations.

Correlation between perturbation of the ionosphere parameters and atmospheric gravity waves generated by tsunamis is investigated in the beginning by Hines (1972) and Peltier & Hines (1976). Then, after the observational techniques were developed, research has been spread to make an effort in establishing possible correlation in detection of ionospheric effects with the earthquake event (Sobolev & Husamiddinov, 1985; Lyperovskaya et al., 2007; Pulinets, 2004). It was suggested by Arai et al. (2011) that it may be possible to indicate tsunami generation by monitoring acoustic-gravity waves in the ionosphere accompanied with undersea seismic disturbances. On the other hand, the earthquake precursor could be related with detection of ionosphere disturbances, observing the formation of ionospheric plasma concentration irregularities (Davies & Baker, 1965). The reason for the possible direct coupling of the processes in the deep earth layers and the ionosphere could be eventual transfer of positive electric charge created by compression in rocks to the layers of ionosphere (Freund et al., 2006). Apart from above mentioned ionospheric perturbation coming from Earth surface, there is a number of them caused by atmosphere or solar activity. There are several studies of the tropical depression influence on ionosphere indicating electrical and electromagnetic effects. One direction of this research is investigation of the sudden disturbances in the low ionosphere which results in changes of radio signals (very low/low frequency (VLF/LF)) that are related to the short-term variations caused by lightning (Price, 2007). In this field, the great advance has been achieved using Global Positioning System (GPS) technology (Erickson et al., 2001) for monitoring ionospheric disturbances during solar flares (Afraimovich, 2000), but also by developing different simulations of the flare effects of the ionosphere (Huba et al., 2005; Meier et al., 2002). As far as ionosphere perturbations caused by GRB are concerned, there are few observational techniques to observe cosmic effects (Nina et al., 2015; Inan et al., 2007).

However, either of these phenomena have influence on the basic ionosphere parameters, such as ion/electron density, electromagnetic filed, pressure and consequently neutral density. The problem with detection of any mentioned parameter is in fact that it is difficult to filter out the origin of the perturbation since the amplitudes of the ionospheric anomalies are usually small. Instead of electromagnetic wave propagation, linear wave theory gives the opportunity to identify and detect frequencies of possible waves propagating within ionosphere (gravity and acoustic modes) but linearization procedure mimic the importance of nonlinear effects on the wave dynamics.

The aim of this paper is to describe perturbation in the ionosphere using compressible fluid model with pressure, rotation, magnetic field and scalar gravitational potential, involving nonlinear terms that are neglected in linear approach. As a result, we obtain conditions for stable vortical structure formation. Simple monitoring of these structures gives an opportunity for fast prediction and reaction of mentioned events that could have influence on humans.

Apart of these advantages, a number of solitary structures are directly observed as elements of plasma motion in the ionosphere and magnetosphere (Hallinan & Davis, 1970), and specially electron and ion density structures on equatorial ionosphere (Lin et al., 2007; Huang et al., 2009).

**It is very clear evidence of the ion density depletion in Fig.1 of (Huang et al., 2009), which can be explained as a consequence of the particles trapped by medium nonlinear scalar potential that will be derived in this paper. One should not expect to observe within the ionosphere vorticies as it is common in the atmosphere, but any regular depletion or enhancement of density is possible to explain by the mechanism that we propose here. Although ionosphere is very**

complex and dynamical, there is number of approximation used in previous theoretical and numerical research that worked well comparing with observed parameters. For example, there is an interaction between sheared zonal flow within ionosphere and Rossby solitons created in the atmosphere of the Earth, making a turbulent stage via accumulation of the flow energy into vortical structures. Here, we propose existence of the magnetized stable vorticies created in the ionosphere that could be used as a transient stage of highly turbulent medium. Recent explanations of the zonal flow creation are due to innhomogenous heating of the atmospheric and ionospheric layers by solar radiation, but previously, there is number of literature investigating the creation of sheared zonal flow as a consequence of nonlinear mechanism excitation by planetary waves or tides. As suggested by Immel et al. (2006), it is not unexpected that tides are able to modulate the dynamo electric fields within the E and F-layer. It would be of great interest to investigate how this interaction influences creation of magnetized solitons derived in this research, since the model of the ionosphere includes conductive fluid with coupled strong interaction of charged particles via Lorentz force. Since the ionosphere represents a slightly ionized gas, so that neutral particles are dominant component, it is expected that self-gravity in the horizontal plane would play significant role in the structure formation. It is more realistic to use scalar self-gravity potential accompanied with Poisson's equation and treat the fluid as compressible, instead of incompressible fluid description using stream function. According to Haldoupis & Pancheva (2002), the conviction of the horizontal plasma transport as being unimportant in E-layers (since the scales involved are much larger than the vertical ones), had to be reconsidered due to new evidence, by Tsunoda et al. (1998), which suggested a link between E-layers and planetary waves. Planetary waves are global scale oscillations in neutral wind, pressure, and density, which prevail and propagate zonally in the mesosphere and lower thermosphere and have periods mostly near 2, 5, 10, and 16 days (Forbs & Leveroni, 1992). It has been shown that horizontal formations capture medium particles and transfer these particles in their movement. Here we show that, as analogy of planetary waves, there are nonlinear wave formations in the ionosphere that could be used in order to follow the dynamics of the ionosphere in easier manner. Therefore, the vortices can considerably contribute in the convective intermixing of a medium. Also, there is a number of simulations that have investigated different processes within the ionosphere that are possible to interprete by the nonlinear solitary solution, e.g. Maruyama et al. (2016) which discusses the density peak structure. As far as the experimental confirmation of the rotation importance on the soliton creation is concerned, we recommend the work of van Hejist & Kloosterziel (1989).

## 2 Ionosphere model: basic equations and approximations

### 2.1 Basic equations

Analytical solution of set of nonlinear partial differential equations, if possible, would give better insight in different processes that are responsible for creation and distortion of such structures by deriving and investigating conditions for their existence. Although there are plenty of papers considering similar topic (Kaladze, 1998; Kaladze et al., 2004; Khantadze et al., 2009), in all of them has been used assumption that is consequence of general gravitational potential action.

In this paper, we use the general scalar gravitational potential, together with Poisson's equation, instead of stream function or shallow water assumption, in order to derive conditions for ionosphere perturbation to take the shape of soliton vortex. It is analyzed in details the condition for the soliton existence and shape within the ionosphere at low latitudes, close to equator.

We assume the ionosphere as a fluid consisting of neutral and charged particles, with $z$ as coordinate measuring distance from the Earth's surface to the two dimensional plain surface in the ionosphere. Since the ionosphere fluid is ionized with the ionization degree depending on the distance from the Earth's surface, there are defined three layers D, E and F, where each of them contains charged particles (electrons and ions) and neutrals. The neutral gas is strongly influenced, via the collisional coupling with low-density ions and electrons, by electromagnetic force. So that, in the momentum equation for the neutral gas exists, apart of Coriolis force, pressure and gravitation, ion-neutral and electron-neutral collisional drag force, via electromagnetic force. We have neglected the inclination of geomagnetic and Earth's north pole of $11^0$ just for the simplicity, with no loose of generality.

As far as the gravitational force is concerned, the ionosphere is influenced by the Earth's gravitation in vertical direction but, for the first time here, we add the Poisson's equation for gravitational potential of the neutral gas, relevant for this geometry, on contrary to usual approach based on assumption of shallow water theory (Kaladze et al., 2004), or using stream function description for incompressible fluid (Kaladze, 1998). We use finite thickness approximation in order to estimate gravity influence on the ionospheric gas dynamics, not only in vertical direction, but mainly in horizontal plain, relevant for vortex soliton formation (Vukcevic, 2019). Assumption of shallow water theory is just a consequence of general Poisson's equation, approximated in horizontal plain, and it will be shown in this work. The set of closed system of equations describing the ionosphere reads as: continuity equation for compressible fluid,

$$\frac{\partial \rho}{\partial t} + \nabla \cdot (\rho \boldsymbol{v}) = 0, \tag{1}$$

where $\rho$ is neutral gas volume density, $\boldsymbol{v}$ is neutral gas velocity;

equation of motion,

$$\frac{\partial \boldsymbol{v}}{\partial t} + (\boldsymbol{v} \cdot \nabla)\boldsymbol{v} + 2(\boldsymbol{\Omega} \times \boldsymbol{v}) + \frac{1}{\rho}(\boldsymbol{j} \times \boldsymbol{B_0}) = \nabla \Phi + \frac{1}{\rho}\nabla P, \tag{2}$$

where $\boldsymbol{\Omega}$ is the angular velocity of the Earth's rotation, , $\boldsymbol{j}$ is the conduction current density, $\boldsymbol{B_0}$ is the geomagnetic field, $\Phi$ and $P$ are scalar three-dimensional gravitational potential and pressure, respectively;

Poisson's equation

$$\Delta \Phi = -4\pi G \rho, \tag{3}$$

with $\rho$ and $\Phi$ previously defined;

current equation,

$$\boldsymbol{j} = \sigma_E \boldsymbol{E} + \frac{\sigma_E}{B_0}(\boldsymbol{B_0} \times \boldsymbol{E}) = en(\boldsymbol{v} - \boldsymbol{v_e}), \tag{4}$$

where $\boldsymbol{E}$ is dynamo electric field, $\sigma_E$ is conductivity tensor, $n$ is number density of charged particles, $e$ is electron charge and $\boldsymbol{v_e}$ is electron velocity;

electric dynamo filed equation,

$$\boldsymbol{E} = (\boldsymbol{v} \times \boldsymbol{B_0}). \tag{5}$$

Here we use following plasma condition in the ionosphere: ions are considered as unmagnetized, so that $\boldsymbol{v_i} = \boldsymbol{v}$, ion velocity across the magnetic field is equivalent to gas velocity and ions are dragged by neutral gas motion completely, while the electrons are magnetized, frozen in the external magnetic filed, so that $\boldsymbol{v_e} = (\boldsymbol{E} \times \boldsymbol{B_0})/B_0^2$ (Kaladze et al., 2004). **Details of derivation of Eq. (2) and involvement of electric field are given in Appendix A. Eq. (2) is the same as Eq. (A5).** In the equation of motion, viscous effects are neglected due to high Hartmann number for typical ionosphere parameters ($Ha^2 = \frac{\sigma B_0^2 L^2}{\eta \rho} \sim 10^5$),
where $\eta$ is kinematic viscosity $\cong 10^{-5} kg/ms$ (Kaladze et al., 2004).

In this research, scalar gravitational potential is related to neutral gas at the $z = z_0$ distance from the Earth surface, where it is defined by the Earth's gravitation only in the $z$ direction but remains the two-dimensional, horizontal component defined by gas in the vicinity of fixed distance from the Earth. Assumption of the stratified stable ionospheric layer involves the Brunt-Väisälä frequency which is fast comparing to large-scale horizontal motion that will be considered here using drift
approximation. We will show that the general scalar gravitational potential is equivalent to the effective height of shallow water theory within approximation of Poisson's equation as it was proposed by Vukcevic (2019). Within that approximation, scalar potential is evaluated as two-dimensional denoted by $\phi$, while volume density is evaluated by surface density $\sigma$ and pressure as two-dimensional pressure denoted by $p$. Details of Poisson's equation approximation are given in Appendix B.

### 2.1.1 Drift approximation

In order to qualitatively estimate contribution of rotation, gravity, pressure and magnetic effects we will employ drift approximation, and as first, we may assume a pseudo-three-dimensional case such that

$$\frac{\partial \boldsymbol{v}}{\partial z} = \epsilon(\nabla_\perp \cdot v_\perp), \tag{6}$$

which is in a good agreement with experimental data (Dokuchaev, 1959).

Here, the subscript $\perp$ indicates the components of the variables within the ionosphere plane surface, and $\epsilon$ is small parameter
of the order of either $(2b\Omega)^{-1}\frac{d}{dt}$ or $(2b(\Omega + enB_0/\rho))^{-1}\frac{d}{dt}$, where $b$ is corrective factor, denoting $\sin\varphi$, and $\varphi$ is latitude measured from the equator.

Last assumption is consistent with the condition of existence of a drift wave and physically means that fluid inertia in the direction of the ambient rotation or/and magnetic field is negligible, or equivalently, ionospheric motions in vertical direction, defined by $z$, are much less than those in horizontal one, defined by $x$ and $y$ (Pudovkin, 1974).
Within a local Cartesian system defined by $\boldsymbol{e_x}$, $\boldsymbol{e_y}$ and $\boldsymbol{e_z}$ as east, north and up direction, respectively (see Fig.1), Earth's angular velocity has following components $\boldsymbol{\Omega} = \Omega(0, \sqrt{1-b^2}\Omega, b\Omega)$, where the equator is defined by $b = 0$, while the pole is defined by $b = 1$. Consequently, geomagnetic field, assumed as a magnetic dipole, has components as $\boldsymbol{B_0} = B_0(0, \sqrt{1-b^2}B_0, -2bB_0)$.

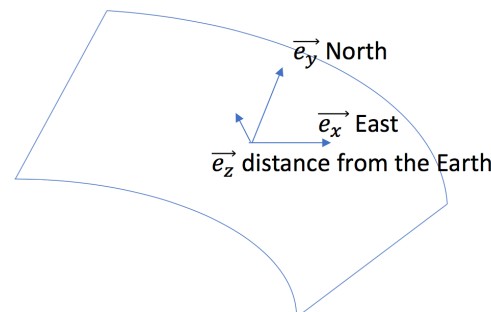

**Figure 1.** Local coordinate system. In horizontal plane are defined two axises by unit vectors: $\boldsymbol{e_x}$ oriented to the east and $\boldsymbol{e_y}$ oriented to the north. Vertical axis to the horizontal plane is defined by unit vector $\boldsymbol{e_z}$; along this axis is defined distance of the horizontal plain from the Earth surface.

Making a vector product of equation of motion, Eq. (2), and $\boldsymbol{e_z}$, we obtain

$$(\frac{\partial \boldsymbol{v}}{\partial t} + (\boldsymbol{v} \cdot \nabla)\boldsymbol{v}) \times \boldsymbol{e_z} + 2(\boldsymbol{\Omega} \times \boldsymbol{v}) \times \boldsymbol{e_z} + \frac{1}{\rho}(\boldsymbol{j} \times \boldsymbol{B_0}) \times \boldsymbol{e_z} = (\nabla\phi + \nabla p) \times \boldsymbol{e_z}. \tag{7}$$

Let us now investigate in details second and third term in the lefthand side of Eq. (7) denoting them as $f_R = 2(\boldsymbol{\Omega} \times \boldsymbol{v}) \times \boldsymbol{e_z}$ and $f_H = \frac{1}{\rho}(\boldsymbol{j} \times \boldsymbol{B_0}) \times \boldsymbol{e_z}$. $f$ represents coupled rotational and magnetic filed contribution. Writing them as

$$f = f_R + f_H = 2b(\Omega + \frac{enB_0}{\rho})\left[v_x, v_y, 0\right] - \frac{2enbB_0}{\rho}\left[5(1-b^2)v_x, (1-2b^2)v_y, 0\right]. \tag{8}$$

we are able to derive relevant parameter necessary for applying drift approximation. In order to compare these two terms in the last expression, rotation and magnetic field contribution, we next consider two extreme cases, pole and equator.

Case (a): In the extreme case $b = 1$, at the pole, the Eq. (8) is simplified and reads

$$f = 2(\Omega - \frac{enB_0}{\rho})\left[(v_x, v_y, 0)\right]. \tag{9}$$

This result is similar with the result obtained by Kaladze (1998) but with different approach of stream function for incompressible fluid, with no gravitation involved, for motions far from the equator. In that approach, used, so called $\beta$ plane approximation, breaks down at polar latitudes.

For very high latitudes, close to pole, $b \to 1$, Eq. (8) becomes

$$f = f(x,y) = 2\Omega\left[(v_x, (1 - \frac{1.5enB_0}{\Omega\rho})v_y, 0)\right]. \tag{10}$$

Ratio of magnetic filed and rotation defines $y$ component of velocity, conditioning the shape of structure. We will discuss this result and implications on the solution in the Section 4.

Case (b): In the extreme case, at the equator plane $b = 0$, structure formation is not possible since both terms in Eq. (8) are equal to 0.That result is the same as obtained in number of papers considering the same problem (Kaladze, 1998). Let us investigate the case for low latitudes, close to equator, $b \to 0$, since there are plenty of observed structures within that region, that has no explanation at all. The Eq. (8) for latitudes $\theta \gtrsim 6^0$ becomes

$$f = f(x,y) = 0.2\Omega \left[ (1 - \frac{5enB_0}{\Omega\rho})v_x, v_y, 0 \right]. \tag{11}$$

For such case, existence of the ambient magnetic filed would modify the solution depending of ratio of magnetic filed and rotation contribution.

## 3  Non-linear equation

Applying drift approximation and using Poisson's equation approximated by two-dimensional functions for density and scalar potential in order to derive non-linear equation, set of equations (1)-(3) will transform to

$$\begin{cases} \dfrac{\partial}{\partial t}\sigma + \{\phi, (\sigma_0 + \sigma)\} = 0, \\[2mm] \sigma = -A\nabla_\perp^2 \phi + B\phi. \end{cases} \tag{12}$$

Here the drift velocity is defined by

$$\boldsymbol{v}_d = \nabla\phi \times \boldsymbol{e}_z \tag{13}$$

while $\boldsymbol{v}_i$ is the inertial velocity, and it depends on the velocity given by Eq. (7). Three-dimensional Poisson's equation is evaluated in two-dimensional plane geometry, in the neighborhood of $z = z_0$ as proposed by Vukcevic (2019), involving thickness of the plain via functions $A$ and $B$.

Case (a): At the pole, inertial velocity is defined by

$$\boldsymbol{v}_i = (\frac{\partial}{\partial t} + \boldsymbol{v}\cdot\nabla)\boldsymbol{v} + (2\Omega + \frac{enB_0}{\rho})\boldsymbol{v}. \tag{14}$$

In the limit of low-frequency perturbations (which is equivalent to long period perturbation; according to observations solitary structure lasts from few hours up to few days, in size of few tens up to few kolometers (Lin et al., 2007; Anderson et al., 2002)) we can omit inertial terms in the further calculations, so that velocity is approximated by $\boldsymbol{v}_d$. It will result in the normalization of the variables by the factor $2\Omega + H$, where $H = \frac{enB_0}{\rho}$. After this assumption all variables will be evaluated within the $(x,y)$ plain, and Poisson's equation is approximated with the finite thickness evaluation in the vicinity of $z = z_0$, which is at the certain distance from the Earth's surface (for details, see Appendix B).

Consequently, looking for the stationary waves which are described by Eq. (12) assuming that $\phi = \phi(y - ut, x)$, where $u$ is a constant parameter meaning the wave velocity along $y$. $x$ and $y$ are local coordinates previously defined. Then, Eq. (12) takes the form

$$
\begin{aligned}
-2u(\Omega + H)\frac{\partial}{\partial y}(B\phi - A\nabla^2\phi) - \frac{1}{2}B'\frac{\partial}{\partial y}\phi^2 + A(\nabla\phi \times \nabla)_z\nabla^2\phi + \\
+ (\phi'_0 B - \sigma'_0)\frac{\partial}{\partial y}\phi - \phi'_0 A\frac{\partial}{\partial y}\nabla^2\phi = 0,
\end{aligned}
\tag{15}
$$

where $'$ denotes derivative with respect to $x$. Last equation corresponds to the Eq. (B13) in the Appendix B, if divided by $2(\Omega + H)$, which represents $f$ in this case.

If the latitudes are close to pole, due to change in inertial velocity, the shape of solution will be changed because the $y$ component of the velocity depends on the ratio of magnetic and rotation values. The shape of solution will be discussed and given in next section.

Case (b): For latitudes close to equator, the second term in the expression for inertial velocity is conditioned by values of magnetic filed influence and Coriolis force influence (see Eq. (8) and (11)). Consequently, it influences normalization value for the velocity, as well as the shape of the soliton. If the magnetic field strength is small compared to rotation, nonlinear equation is similar as previous equation and reads as

$$
\begin{aligned}
-0.2uf\frac{\partial}{\partial y}(B\phi - A\nabla^2\phi) - \frac{1}{2}B'\frac{\partial}{\partial y}\phi^2 + A(\nabla\phi \times \nabla)_z\nabla^2\phi + \\
+ (\phi'_0 B - \sigma'_0)\frac{\partial}{\partial y}\phi - \phi'_0 A\frac{\partial}{\partial y}\nabla^2\phi = 0.
\end{aligned}
\tag{16}
$$

Consequently, soliton shape is symmetric and the amplitude of the soliton is changed. In the case when magnetic parameter $H$ is order of $0.2\Omega$ solution will be elongated along $y$ axis while if $H \geqslant 0.4\Omega$ soliton will change moving direction and one can expect structure elongated along the $x$ axis. This is because normalization value has different velocity components and these two cases will be estimated and discussed in next section.

Here, we underline difference between the Rossby waves derived using so-called geostrophic approximation for number of fluids, planetary atmospheres or plasma drift waves (Sommeria et al., 1988; Marcus, 1989; Hasegawa et al., 1979), and nonlinear soliton wave solution discussed by Petviashvili (1983) and Vukcevic (2019). The first reason for different structure comes from different dispersion relations derived in linearized problem. In our case it reads as

$$
\omega = \boldsymbol{v_0} \cdot \boldsymbol{k} + \frac{\boldsymbol{k} \cdot (\boldsymbol{e}_z \times \nabla\sigma_0)}{(1 - (A/B)k^2)},
\tag{17}
$$

while in the case for Rossby waves it is

$$
\omega = \frac{\boldsymbol{k} \cdot (\boldsymbol{e}_z \times \nabla\sigma_0)}{(1 - (A/B)k^2)}.
\tag{18}
$$

Consequently, nonlinear equation describing Rossby waves contains the nonlinear term which is of the vector-type, connected with the term $[\nabla\phi \times \nabla]_z$ (Korchagin & Petviashvili, 1985; Korchagin et al., 1987; Fridman & Khoruzhii, 1999). In our case, crucial term is one connected with term $B'$, which is related with the equilibrium property of the fluid, namely, surface density that is $x$ dependent and and thickness of the layer.

## 4 Results: Soliton vortex solution

In this section it will be shown how the solution of non-linear equation derived in previous section depend on the thickness of the layer. Nonlinear equations (15) and (16) are similar, and we look for the solution of either of them in the form

$$\nabla^2 \phi = \lambda(x)\phi + \nu(x)\phi^2, \tag{19}$$

where $\lambda$ and $\nu$ are functions of $x$ caused by both, an inhomogeneity of the equilibrium functions and thickness of the plain surface, that we need to find out. Functions $\lambda$ and $\nu$ read as follows:

$$\lambda(x) = \frac{1}{A}(B - \frac{\sigma_0'}{u}), \tag{20}$$

$$\nu(x) = \frac{1}{2u}(\lambda A' + \frac{\sigma_0''}{u}), \tag{21}$$

where $''$ represents second derivative with respect to $x$.

Even for constant $B$ ($B' = 0$), nonlinear term $\phi^2$ remains due to gradient of factor $A$ and the extremum of the function $\sigma_0$, which is equilibrium property of the surface density. It is important result since it suggests that equilibrium thickness of the layer cannot be taken as constant; thickness is related either with gradient of $A$ or with surface density function extremum (see Eq. 21).

Then, the stable solution of Eq. (15) or (16) reads:

$$\phi = \frac{2\lambda}{\nu}F(R), \tag{22}$$

where $R = \sqrt{\lambda}r$ is the dimensionless radius in the moving frame, and $F$ is the solution of the equation

$$\frac{1}{R}\frac{\partial}{\partial R}R\frac{\partial f}{\partial R} = F - F^2. \tag{23}$$

The approximate solution of the Eq. (23) is (Zakharov & Kuznetsov, 1974)

$$F = 2.4(\cosh(\frac{3}{4}R))^{-\frac{4}{3}}, \tag{24}$$

which means that potential has solution taking the form of steady solitary vortex shown in Fig. 2, where $R$ represents dimensionless distance to the center of the vortex. Vortex is traveling along $y$ coordinate, northward in our description, with constant velocity $u$.

Let us now investigate the cases when the normalization value is not symmetric. Close to pole, if the magnetic parameter $H$ is order of $\Omega$ one can expect symmetric solution. If the $H < \Omega$ $v_y$ component will be smaller than $v_x$ (see Eq. (10) and the solution is changed and reads as:

$$F = \cosh(\frac{3}{4}R(1 + f(x)\Lambda))^{-\frac{4}{3}}. \tag{25}$$

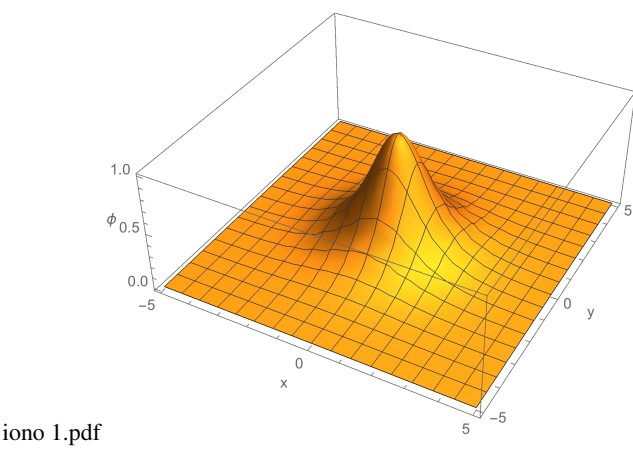

iono 1.pdf

**Figure 2.** Potential derived in Eq. (22). Horizontal axises represent $x$ and $y$ coordinates, while vertical axis represents non-dimensional potential.

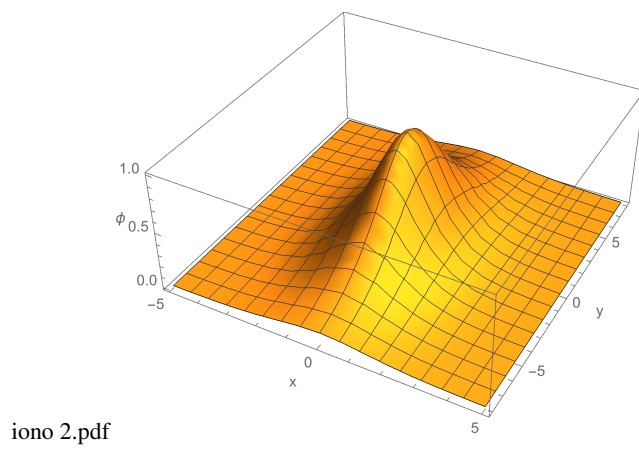

iono 2.pdf

**Figure 3.** Shape of the potential in the equator vicinity elongated along the $y$ axis.

The solution of scalar potential approximated by Eq. (25) represents asymmetric soliton vorticity shape of scalar potential shown in Fig. 4.

Related to these two types of potential given by Eq. (24) and Eq. (25), there are three different possibilities in the area close to equator: $H \ll \Omega$ solution is symmetric with the amplitude higher than in the symmetric case close to pole; $H \sim 0.2\Omega$ solution will be elongated along the $y$ axis and it is shown in Fig. 3; if $H \geqslant 0.4\Omega$ soliton will be symmetric but it will change the moving direction, since the $v_x$ component has opposite sign.

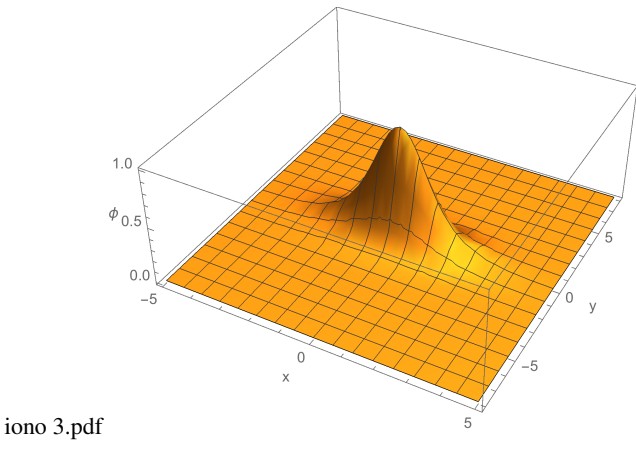

iono 3.pdf

**Figure 4.** Shape of the potential close to pole elongated along the $x$ axis.

## 5   Discussion

In this work we have found soliton solution in few different cases, on the pole, close to pole and close to the equator of the ionosphere. We have confirm few results obtained by using different approach with emphasis on the cause of the obtained phenomena. It is shown that the surface mass density disturbance in shallow water theory approach is just consequence of scalar gravitational potential action via Poisson's equation used in this work. Also, we have proved that the thickness of the layer can be uniform (see Eq. (21), even if $A' = 0$ remains the term $\sigma''_0$) but the soliton existence is driven by the inhomogeneity of the equilibrium mass surface density. The advantage of such solution in studying the Ionospheric Disturbances of any kind, is the fact that the amplitude and velocity of the soliton are very sensitive to ionospheric parameters (density of charged particles, and consequently of pressure, gravity, magnetic filed and rotation velocity values), which means that once the balance between all of them is failed, it will result in the distortion of the structure. Thus, it is very easy to predict consequences caused by maximum or minimum neutral, so that electron concentration, just by simple monitoring of the structure dynamics within the ionosphere region. It is on contrary to the investigation of the ionospheric anomalies influence by some other methods, involving very small amplitudes of either linearized periodic waves or electromagnetic waves that need to be filtered out in order to define the source, after number of reflection by ionosphere.

It is important to underline that the type of the vortex, when it is symmetric, is defined by the extremum value of density inhomogeneity via term in $\nu$ Eq. (21): cyclones by minimum, anticyclones by maximum. Also, if the potential is bright soliton, consequently the density is represented by dark one since particles are trapped by such potential, and vice versa, whenever the potential is dark soliton, density will be enhanced, represented by bright one. It is possible further to investigate more detailed conditions of the solitary structure for different ionosphere layers.

(I) Ionosphere D-layer: Within this region, placed at the $50 - 100$ km from the Earth's surface, we can assume that contribution of charged particles can be neglected, so the ponderomotive force effects are small compared to Coriolis force effects (Gershman, 1974). It means that it is likely to expect solitary structure for small latitudes, close to equator, elongated along the

$y$ coordinate, while for high latitudes and at the pole, the soliton is symmetric, the size of the soliton will depend on density gradient and its velocity is normalized by $f = f_R = \Omega$.

(II) Ionosphere E-layer: This layer is placed at the $100 - 150$ km from the Earth's surface, and one can expect creation of soliton at all latitudes higher than $6^0$, since the ponderomotive force is of the order of the Coriolis one. In this case, soliton velocity is defined by $f = 2(\Omega + H)$ at the pole and for latitudes close to pole. Since the value $H$ has the opposite sign to $\Omega$, the cancelation of the vortex structure is possible when these two terms are of the same order, or it is possible to change moving direction of the soliton structure. Next, the size of the soliton is defined by $R = \sqrt{\lambda} r$ and consequently, by soliton velocity $u$,

which for this case is defined by $2(\Omega + H)$, one expects the size to increase compared with the same case for D-layer. As far as the low latitude case is concerned, the latitudes close to equator, the size and velocity of the soliton are dependent on the value $H$ compared to $\Omega$, and even more, the soliton is not symmetric but rather extended along the $y$ axes, since $f = f(x, y)$.

(III) Ionosphere F-layer: Ionospheric F-layer is for heights $150 - 400$ km from the Earth's surface. In this case, for all latitudes soliton structure is mainly defined by value $0.2H$. At the pole, the soliton is symmetric, velocity is defined by $f = 2H$, while

close to equator one can expect soliton elongated along the $x$ axis but moving in opposite direction compared to E and D-layer because $f = f(x, y) = f_H$.

That nonlinear effects are extremely important in wave dynamics whenever they are comparable with dispersive properties of the medium, that are defined by gradients of pressure and gravity potential, rotation and magnetic force. One very good example of the balance between nonlinear and dispersive effects is tsunami, surface gravity wave that can propagate for a long

time with large constant velocity and large amplitude, with no distortion or dissipative effects, as long as the conditions of balance are satisfied. So that, such solution is stable in space and time, propagating with the constant velocity and amplitude. The fact that this solution is very sensitive to even very small change in any parameter, makes possible instant detection of such a change by distortion of the structure.

## 6    Conclusions

We have studied non-linear dynamics of fine structures in the Earth's ionosphere within drift approximation, with primary emphasis on necessary conditions for vortex soliton creation. We have included all relevant physical forces in the model, keeping nonlinear terms, and applied it at the equator of the ionosphere. A series of direct observation of such soliton structures are carried out either from the Earth's surface or on board the satellites. We have summarized all possible soliton structure formations at different latitudes, as well as at different ionospheric layers. The soliton size and velocity are constant but

defined by different values of ionospheric parameters. From our investigation of the soliton solution in the conditions of Earth ionosphere we can outline following conlusions:

    – Stable localized solution of partial nonlinear differential equation is possible under the balance between nonlinear terms and dispersion, while dispersion is caused by Coriolis force and magnetic filed force on one side, and gravity and pressure on the other side. Necessary condition for its existence is either nonuniform thickness of the layer or existence

of extremum of equilibrium mass surface density function.

- The amplitude and velocity of the soliton are very sensitive to ionospheric parameters (density of neutrals, charged particles, and consequently of pressure, gravity, magnetic filed and rotation velocity values), which means that once the balance between all of them is failed, it will result in the distortion of the structure.

- In general, nonlinear equation that has soliton solution is possible at all latitudes higher than $6^0$ but the physical processes responsible for it are different. Close to equator, presence of magnetic field is crucial since effects of rotation are very small, while on the pole it is always combination of the rotation and magnetic effects, with the possibility for the soliton to vanish due to opposite signs of these two effects.

- Soliton existence is direct consequence of the equilibrium condition on the layer thickness. The equilibrium must be defined either by the extremum of surface density function or by parameter $A$.

- In the ionosphere D-layer it is likely to expect solitary structure close to equator, but if it is created soliton is elongated along the $y$ coordinate. Close to pole, the size of the soliton will depend on density gradient and its velocity is normalized by $\Omega$.

- Ionosphere E-layer is characterised by two types of the solitons: close to pole soliton vortex is symmetric, with size larger then in D-layer with the possibility to vanish, and close to equator the existence is caused by the magnetic field presence, and the soliton is elongated along $y$ axes.

- Opposite situation is within the ionosphere F-layer where at the pole exists a symmetric soliton, larger then in the ionosphere D-layer at the pole, while close to equator exists extended soliton structure.

Finally, we hope that this model will be used in explanation of the ionosphere structures, as well as in testing physics background of complex ionosphere simulations. This model can be used not only to model ionosphere structure, but also for different astrophysical systems as eg. accretion disks, where the thickness effects could be very important. Therefore, finite thickness effects should be taken into account. However, this approach can be improved trying to find out correlation between soliton structure dynamics and other methods used to identify the ionospheric anomalies. Also, it would be of great importance to investigate the stability of soliton structure as subject of small disturbance and apply it on the study of interaction between the solitons within different ionospheric layers. All of these mentioned issues will be consider in further research.

**Appendix A: Derivation of the momentum equation for the neutral gas influenced by low-density ions and electrons**

As it is mentioned in Section 2, the neutral gas is strongly influenced by strong electric field as interaction of coupled charged particles, electrons and ions. That interaction enters in the neutral gas momentum equation via collisional drag forces of neutral particles and charged particles in the ionosphere. So that, there are three coupled momentum equations

$$\frac{\partial \boldsymbol{v}}{\partial t} + (\boldsymbol{v} \cdot \nabla)\boldsymbol{v} + 2(\boldsymbol{\Omega} \times \boldsymbol{v}) = \nabla \Phi + \frac{1}{\rho}\nabla P + m_i n_i N <\Sigma \nu>_{in} (\boldsymbol{v_i} - \boldsymbol{v}) + m_e n_e N <\Sigma \nu>_{en} (\boldsymbol{v_e} - \boldsymbol{v}), \tag{A1}$$

$$340 \quad m_i n_i \frac{\partial \boldsymbol{v_i}}{\partial t} = \nabla P_i + n_i e_i (\boldsymbol{E} + \boldsymbol{v_i} \times \boldsymbol{B}) - m_i n_i N < \Sigma\nu >_{in} (\boldsymbol{v_i} - \boldsymbol{v}), \tag{A2}$$

$$m_e n_e \frac{\partial \boldsymbol{v_e}}{\partial t} = 0 = \nabla P_e - n_e e (\boldsymbol{E} + \boldsymbol{v_e} \times \boldsymbol{B}) - m_e n_e N < \Sigma\nu >_{en} (\boldsymbol{v_e} - \boldsymbol{v}). \tag{A3}$$

In these three equations index i/e denotes ion and electron, respectively, while $< \Sigma\nu >_{in}$ and $< \Sigma\nu >_{en}$ are the collision cross sections for neutrals with charged particles. Consequently, collision frequencies are $\nu_i = N < \Sigma\nu >_{in}$ and $N < \Sigma\nu >_{en}$ for the ions and electrons. Coriolis force for ions and electrons is negligible compared to the Lorentz force since

$\nu_e \sim 10^6 1/s >> \omega_{ci} \sim 2 \times 10^2 1/s$, and $\nu_i \sim 10^3 1/s << \omega_{ce} \sim 6 \times 10^6 1/s$, both are much higher than the horizontal component of Coriolis acceleration $f_c \sim 6 \times 10^{-5} 1/s$ at midlatitude for E layer, where $\omega_{ci,e} = eB/m_{i,e}$ are the cyclotron frequencies for ions and electrons, respectively. From the same estimation it is obvious that ion frictional force is much higher than electron one due to mass ratio of electron and ion. Adding Eq. (A2) and (A3) we obtain

$$m_i n_i \frac{\partial \boldsymbol{v_i}}{\partial t} = \nabla (P_i + P_e) + n_e e (\boldsymbol{v_i} - \boldsymbol{v_e}) \times \boldsymbol{B} - m_i n_i N < \Sigma\nu >_{in} (\boldsymbol{v_i} - \boldsymbol{v}) - m_e n_e N < \Sigma\nu >_{en} (\boldsymbol{v_e} - \boldsymbol{v}), \tag{A4}$$

where it has been used quasy-neutrality condition $n_i = n_e = n$. Applying $\boldsymbol{v_e} = \boldsymbol{E} \times \boldsymbol{B}/B^2$ from the Eq. (A3) in Eq. (A4), neglecting higher order drift contribution, such as diamagnetic drift and since $\boldsymbol{v_i} = \boldsymbol{v}$ due to very rapid ion velocity evolvement, using $n_e e (\boldsymbol{v} - \boldsymbol{v_e}) = \boldsymbol{v} = \boldsymbol{j}$, we substitute Eq. (A4) into Eq. (A1) and we finally obtain momentum equation for neutral gas as following

$$\frac{\partial \boldsymbol{v}}{\partial t} + (\boldsymbol{v} \cdot \nabla)\boldsymbol{v} + 2(\boldsymbol{\Omega} \times \boldsymbol{v}) = \nabla\Phi + \frac{1}{\rho}\nabla P + \frac{1}{\rho}(\boldsymbol{j} \times \boldsymbol{B}). \tag{A5}$$

In the last equation we have neglected ion and electron pressure comparing with neutral gas pressure due to $P_{e,i}/P \sim n/N << 1$. The electron Hall current contribution is a driving term from the ionospheric electric field $\boldsymbol{E}$ since Hall conductivity $\sigma_{EH}/B$ is much higher than Pedersen conductivity $\sigma_{EP} \sim \sigma_{EH}\omega_{ci}/\nu_i$ due to fact that for E layer typical conditions $\omega_{ci}/\nu_i << 1$, ions could be considered to be unmagnetized. Electrons are magnetized and frozen in the external magnetic field experiencing only drift perpendicular to the magnetic field. That is why the parallel conductivity is high $\sigma_{E||} \sim \sigma_{EP}\omega_{ce}/\nu_e$

because $\omega_{ce}/\nu_e >> 1$, ending with equation for electric current as

$$\boldsymbol{j} = en(\boldsymbol{v} - \boldsymbol{v_e}), \tag{A6}$$

known as noninductive approximation (for more details see chapter 7, specially 7.2 of Schunk & Nagy (2009)). Eq. (A5) is the same as Eq. (2) in the main text.

## Appendix B: Finite thickness approximation

We have restricted problem studying the two-dimensional motion on the horizontal surface Eq. (6), therefore we must solve three-dimensional Poisson's equation for a two-dimensional geometry. To develop an analytical theory, we need an appropriate approximation of Eq. (3).

We assume that the three-dimensional potential and density may be written, in the neighborhood of $z = 0$ in the forms of

$$\Phi(r,\theta,z) = \phi(r,\theta)f(z), \tag{B1}$$

$$\rho(r,\theta,z) = \sigma(r,\theta)g(z). \tag{B2}$$

Integrating last equation with respect to $z$, we obtain

$$-A\nabla_\perp^2 \phi + B\phi = \sigma, \tag{B3}$$

where $\nabla_\perp^2$ is the two-dimensional Laplacian in the horizontal surface , $A = \frac{a}{c}$, $B = \frac{b}{c}$

$$a = \int_{-D}^{D} f(z)dz, \tag{B4}$$

$$b = \int_{-D}^{D} f''(z)dz, \tag{B5}$$

and

$$c = -\int_{-D}^{D} g(z)dz. \tag{B6}$$

The conventional Lin-Shu approximation for infinitely thin disk corresponds to the limit of $B = 0$ which is obtained for $g(z) = \delta(z)$. All details about the normalization of the variables and separation of ambient and fluctuation parts are given in the Vukcevic (2019). In order to obtain Eq. (15) we start from Eq. (12) in the following form

$$\frac{\partial \sigma}{\partial t} + \nabla \cdot (\sigma \boldsymbol{v}) = 0, \tag{B7}$$

where $\boldsymbol{v} = \nabla \phi \times \boldsymbol{e_z}$ and $\sigma = -A\nabla^2 \phi + B\phi$. Second term of the Eq. (A1) reads

$$\nabla \cdot (\sigma \boldsymbol{v}) = \boldsymbol{v_0}\nabla\sigma + \boldsymbol{v}\nabla\sigma_0 + \boldsymbol{v}\nabla\sigma, \tag{B8}$$

since $\nabla \boldsymbol{v} = \nabla \boldsymbol{v_0} = 0$. Then, each term will be

$$\boldsymbol{v_0}\nabla\sigma = (\nabla\phi_0 \times \boldsymbol{e_z}) \cdot \nabla(-A\nabla^2\phi + B\phi) = A\phi_0' \frac{\partial}{\partial y}\nabla^2\phi - B\phi_0'\frac{\partial\phi}{\partial y}, \tag{B9}$$

$$\boldsymbol{v}\nabla\sigma_0 = (\nabla\phi \times \boldsymbol{e_z})\frac{\partial}{\partial x}\sigma_0(x) = \sigma_0'\frac{\partial\phi}{\partial y} \tag{B10}$$

$$\boldsymbol{v}\nabla\sigma = (\nabla\phi \times \boldsymbol{e_z}) \cdot \nabla(-A\nabla^2\phi + B\phi) = -A'\frac{\partial\phi}{\partial y}\nabla^2\phi - A(\nabla\phi \times \nabla)\nabla^2\phi + B'\phi\frac{\partial\phi}{\partial y}. \tag{B11}$$

Collecting all these terms, neglecting the first term in last equation as a higher order nonlinear term, Eq. (B1) reads as

$$\frac{\partial}{\partial t}(-A\nabla^2\phi + B\phi) + \frac{\partial\phi}{\partial y}(\sigma_0' - B\phi_0') - A(\nabla\phi \times \nabla)\nabla^2\phi + \phi_0'A\frac{\partial}{\partial y}\nabla^2\phi + \frac{1}{2}B'\frac{\partial\phi^2}{\partial y} = 0. \tag{B12}$$

In order to show that solitary vortex solution satisfies the last equation, we assume solution to be function $\phi = \phi(y - ut, x)$, so that derivative with respect to time will become $\frac{\partial}{\partial t} = -u\frac{\partial}{\partial y}$, and consequently Eq. (B6) will read as

$$-u\frac{\partial}{\partial y}B\phi + uA\frac{\partial}{\partial y}\nabla^2\phi + \frac{\partial\phi}{\partial y}(\sigma_0' - B\phi_0') - A(\nabla\phi \times \nabla)\nabla^2\phi + \phi_0'A\frac{\partial}{\partial y}\nabla^2\phi + \frac{1}{2}B'\frac{\partial\phi^2}{\partial y} = 0, \tag{B13}$$

or

$$A(\phi_0 + u)\frac{\partial}{\partial y}\nabla^2\phi + \frac{\partial\phi}{\partial y}(\sigma_0' - B(\phi_0' + u)) + A(\nabla\phi \times \nabla)\nabla^2\phi + \frac{1}{2}B'\frac{\partial\phi^2}{\partial y} = 0. \tag{B14}$$

Eq. (B13) is the same as Eq. (15) after normalizing soliton velocity $u$ by $2(\Omega + H)$. If $u >> \phi_0'$ we can further simplify the equation having in the first two terms instead of $(u + \phi_0')$ just $u$. Dividing it by $uA$ one gets

$$\frac{\partial}{\partial y}\nabla^2\phi + \frac{1}{u}(\nabla\phi \times \nabla)\nabla^2\phi - \frac{1}{A}(B - \frac{\sigma_0'}{u})\frac{\partial\phi}{\partial y} + \frac{1}{2uA}B'\frac{\partial\phi^2}{\partial y} = 0. \tag{B15}$$

Next, we show that looking for the solution of that equation in the form

$$\nabla^2\phi = \lambda(x)\phi + \nu(x)\phi^2 \tag{B16}$$

will lead to vanish of the term related with $B'$. Acting by operator $\frac{\partial}{\partial y} + \frac{1}{u}[\nabla\phi \times \nabla]_z$ on all parts of (B10) and neglecting terms of the order $\phi^3$ we get

$$\frac{\partial}{\partial y}\nabla^2\phi + \frac{1}{u}(\nabla\phi \times \nabla)\nabla^2\phi = \lambda(x)\frac{\partial\phi}{\partial y} - \frac{\lambda'}{2u}\frac{\partial\phi^2}{\partial y} + \nu\frac{\partial\phi^2}{\partial y}. \tag{B17}$$

If Eq. (B10) satisfies Eq. (B9) coefficients of corresponding terms must be equal, which gives

$$\lambda(x) = \frac{1}{A}(B - \frac{\sigma_0'}{u}), \tag{B18}$$

and

410 $\quad$ $$\nu(x) = \frac{\lambda'}{2u} - \frac{1}{2uA}B'. \tag{B19}$$

Since

$$\lambda' = \frac{B'A - BA'}{A^2} - \frac{\sigma_0''A - \sigma_0'A'}{A^2 u}, \tag{B20}$$

we have

$$\nu(x) = \frac{1}{2u}\frac{B'A - BA'}{A^2} - \frac{\sigma_0''A - \sigma_0'A'}{2A^2 u^2} - \frac{1}{2uA}B'. \tag{B21}$$

The first and last term give zero, so it becomes

$$\nu(x) = -\frac{uBA' + (\sigma_0''A - \sigma_0'A')}{2u^2 A^2} = \frac{1}{2u}(\lambda A' + \frac{\sigma_0''}{u}). \tag{B22}$$

Details for derivation of nonlinear equation and its solution in the case when inertial velocity is not constant but rather $x$ or $y$ dependent can be found in Appendix B of Vukcevic (2019). In that case drift velocity has to be approximated by $\boldsymbol{v} = \frac{1}{\Omega}\nabla\phi\times\boldsymbol{e_z}$ where $\Omega$ is not constant any more, which implies $\nabla\boldsymbol{v} \neq 0$.

*Author contributions.* M. Vukcevic

*Competing interests.* There os no competing interests

*Disclaimer.* TEXT

*Acknowledgements.* This work is supported by the Ministry of Education and Science of the Republic of Serbia through research project 176001.

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
