# Peer review of "Nonlinear vortex solution for perturbations in the Earth's Ionosphere"

_Nonlinear Processes in Geophysics, 2019_

## Referee Comment (RC1) · Anonymous Referee #1 · 14 Jan 2020

The paper proposes unusual approach for describing large-scale structures in the Earth's ionosphere. I am very surprised by this application of conventions and formalism of neutral atmosphere to plasma environment. Electromagnetic effects are dominant in the ionosphere. Well-studied approximations and corresponding transport equations are described in textbooks. One of the latest and highly regarded is Schunk, R. W., & Nagy, A. F. (2009). Ionospheres: Physics, plasma physics and chemistry (2nd ed.). Cambridge, UK: Cambridge University Press. It is well-accepted that main drivers for the ionosphere-thermosphere (that is a neutral counterpart) are related to space weather and complex electrodynamic coupling with the solar wind and the Earth's magnetosphere. Driving by lower atmosphere (tides, acoustic gravity waves) contributes to the ionospheric dynamics, e.g., Immel, T. J., E. Sagawa, S. L. England,

[Figure]

S. B. Henderson, M. E. Hagan, S. B. Mende, H. U. Frey, C. M. Swenson, and L. J. Paxton (2006), Control of equatorial ionospheric morphology by atmospheric tides, Geophys. Res. Lett., 33, L15108, doi:10.1029/2006GL026161. However, gravitational (and Coriolis force) influence on the ionosphere is negligible compared to electrodynamical processes. Paper by Lin et al. (2007) cited in the manuscript refers to enhancement of F-region dynamo electric field by tidal structures as a plausible mechanism of formation of large-scale features in low-latitude ionosphere. Paper by Huang et al. (2009) cited in the manuscript refers to large-scale plasma and neutral density decreases in the equatorial ionosphere. The authors suggested cooling of the equatorial region as the main cause of the phenomenon and substantiated this explanation by simultaneous temperature measurements. Ionosphere is too dynamic to sustain anything similar to lower atmospheric zonal flows. In my opinion, the manuscript is an interesting theoretical exercise, but I will wait for conclusive observations of vortex structures in the ionosphere.

---

## Referee Comment (RC2) · Anonymous Referee #2 · 17 Jan 2020

This paper presents a theoretical study of small-scale ionospheric perturbations. The authors derive conditions for nonlinear soliton waves in the ionosphere using the MHD description and Poisson's equation. This might be a useful theoretical practice, however, it is unclear how the study can be applied to address outstanding science questions in ionospheric physics. In the conclusion section, the authors present some discussion of solitary structures in different ionosphere layers. Are there any observations of solitary structures in the ionosphere? Observational evidence would be good to mention. In addition, major drivers of the ionosphere, including the solar irradiance, space weather conditions, and lower atmospheric forcing are not considered in the study at all. The electrodynamical effect in the ionosphere is not included either.

[Figure]

2019-56, 2019.

---

## Author Response (AR1)

The paper proposes unusual approach for describing large-scale structures in the Earth's ionosphere. I am very surprised by this application of conventions and formalism of neutral atmosphere to plasma environment. Electromagnetic effects are dominant in the ionosphere. Well-studied approximations and corresponding transport equations are described in textbooks. One of the latest and highly regarded is Schunk, R. W., & Nagy, A. F. (2009). Ionospheres: Physics, plasma physics and chemistry (2nd ed.). Cambridge, UK: Cambridge University Press. It is well-accepted that main drivers for the ionosphere-thermosphere (that is a neutral counterpart) are related to space weather and complex electrodynamic coupling with the solar wind and the Earth's magnetosphere. Driving by lower atmosphere (tides, acoustic gravity waves) contributes to the ionospheric dynamics, e.g., Immel, T. J., E. Sagawa, S. L. England,

S. B. Henderson, M. E. Hagan, S. B. Mende, H. U. Frey, C. M. Swenson, and L. J. Paxton (2006), Control of equatorial ionospheric morphology by atmospheric tides, Geophys. Res. Lett., 33, L15108, doi:10.1029/2006GL026161. However, gravitational (and Coriolis force) influence on the ionosphere is negligible compared to electrodynamical processes. Paper by Lin et al. (2007) cited in the manuscript refers to enhancement of F-region dynamo electric field by tidal structures as a plausible mechanism of formation of large-scale features in low-latitude ionosphere. Paper by Huang et al. (2009) cited in the manuscript refers to large-scale plasma and neutral density decreases in the equatorial ionosphere. The authors suggested cooling of the equatorial region as the main cause of the phenomenon and substantiated this explanation by simultaneous temperature measurements. Ionosphere is too dynamic to sustain anything similar to lower atmospheric zonal flows. In my opinion, the manuscript is an interesting theoretical exercise, but I will wait for conclusive observations of vortex structures in the ionosphere.
* * *
Nonlin. Processes Geophys. Discuss.,
https://doi.org/10.5194/npg-2019-56-RC2, 2020

[Figure]

This paper presents a theoretical study of small-scale ionospheric perturbations. The authors derive conditions for nonlinear soliton waves in the ionosphere using the MHD description and Poisson's equation. This might be a useful theoretical practice, however, it is unclear how the study can be applied to address outstanding science questions in ionospheric physics. In the conclusion section, the authors present some discussion of solitary structures in different ionosphere layers. Are there any observations of solitary structures in the ionosphere? Observational evidence would be good to mention. In addition, major drivers of the ionosphere, including the solar irradiance, space weather conditions, and lower atmospheric forcing are not considered in the study at all. The electrodynamical effect in the ionosphere is not included either.

Answer to Referee# 1

We are grateful to all comments given by Rewiever. We hope that our answers will improve our manuscript and make it more undersrtandable.

1. "I am very surprised by this application of conventions and for- malism of neutral atmosphere to plasma environment. Electromagnetic effects are dominant in the ionosphere. Well-studied approximations and corresponding transport equations are described in textbooks. One of the latest and highly regarded is Schunk, R. W., & Nagy, A. F. (2009). Ionospheres: Physics, plasma physics and chem- istry (2nd ed.). Cambridge, UK: Cambridge University Press."

Answer: We have not applied formalism of neutral atmosphere to plasma environment. We have considered ionosphere as ionized gas with certain degree of ionization; in higher altitudes ionization becomes higher and effects in F region are predominant by electromagnetic force. However, formalism in our manuscript agrees in all steps with the formalism given by mentioned reference: Schunk, R. W., & Nagy, A. F. (2009). Ionospheres: Physics, plasma physics and chemistry (2nd ed.). Cambridge, UK: Cambridge University Press. Namely, we treat one fluid but conducting gas. Interaction between neutrals and charged particles is given by conductivity tensor as result of collision between these particles. Even more, we have discussed separate influence of ions and electrons due to their different mass. If necessary, we can add one more appendix in the manuscript with brief derivation of the equation (2) in the manuscript, which coincide with (7.32) in the reference, after application of the equations (7.44) and after. Simplification of the general MHD equations is necessary because they are rather complicated, as mentioned in the reference.

2. "It is well-accepted that main drivers for the ionosphere-thermosphere (that is a neutral counterpart) are related to space weather and complex electrodynamic coupling with the solar wind and the Earth's magnetosphere. Driving by lower atmosphere (tides, acoustic gravity waves) contributes to the ionospheric dynamics, e.g., Immel, T. J., E. Sagawa, S. L. England…."

   Answer: The aim of the paper is not to discuss any of drivers of the ionosphere but to derive strict analytical solution of simplified but still valid model of ionosphere and to use this type of solution in order to explain and follow up either of mentioned drivers or some phenomena detected within the ionosphere. Also, we have found the condition for the derived solution, in terms that ad hock assumption of the constant thickness of the ionospheric layer can be misused in conclusion. It is similar with Jeans assumption on the constant density for the unstable gas cloud, leading to the condition of the wavelength of instability (and consequently to the frequency) that contradict to the initial assumption. That is why this type of approach has been used.

3. "However, gravitational (and Coriolis force) influence on the ionosphere is negligible compared to electrodynamical processes. "

Answer: This conclusion is not in contradiction what we have derived, but not for all ionosphere long. Equations (8), (9), (10) and (11) are used to discuss influence by rotation and electromagnetic force via exact values of concentration of charged particle n and gas density rho, which can be understood as ionization degree (see discussion I, II and III for each layer separately).

4. "In my opinion, the manuscript is an interesting theoretical exercise, but I will wait for conclusive observations of vortex structures in the ionosphere. "

Answer: We hope that the Referee is assured (answer 2) that this unusual approach is not just for exercise and elegance but rather for re-investigation of the assumption used for many years in number of publications. We underline that the aim of the paper is not to support ionosphere as a whole but to offer 'simple' solution for general probe of the ionosphere dynamics as a control for input parameter in simulations or to explain density drop/ charge particles drop that are observed in many measurements of the ionosphere. Enhancement of the potential given in the figures means that density depletion follows the same shape since particles are trapped within that potential. It confirms even the reference suggested by Referee "L. J. Pax- ton (2006), Control of equatorial ionospheric morphology by atmospheric tides, Geo- phys. Res. Lett., 33, L15108, doi:10.1029/2006GL026161" as well as some others given in the paper: Yumoto K., Ishitsuka M. & Kudeki E. 2002, Geophys. Res. Lett. 29, No. 12, 1596. We have not mentioned many of them as aurora borealis and so.

We hope that Referee will find our answers useful. If there is any more we are here to answer.

Answer to Referee# 2

We are grateful to all comments given by Referee. We hope that our answers will improve our manuscript and make it more understandable.

1. "This might be a useful theoretical practice, however, it is unclear how the study can be applied to address outstanding science questions in ionospheric physics."

Answer:

Study of electromagnetic wave propagation, linear wave theory, gives the opportunity to identify and detect frequencies of possible waves propagating within ionosphere (gravity and acoustic modes) but linearization procedure mimic the importance of nonlinear effects on the wave dynamics.

The best example of neglected nonlinear effects is tsunami wave, wave with constant amplitude and huge constant velocity traveling with no dissipation at all, as long as the condition of balance between nonlinearity and dispersion is assured. Implementation or study of the dynamics using linear waves can mislead to the conclusion that such waves would break down and rise up in shorter time scale, due to dispersive properties of the medium. If the medium is dispersive and nonlinearity can balance it, then it can break down after traveling long distance, gaining a huge amount of energy that is realized suddenly after the balance is broken. If predicted earlier, it can safe either human lives or electric devices.

Simple monitoring of these structures gives an opportunity for fast prediction and reaction of mentioned events that could have influence on humans. In our research, we have identified parameter for dispersion and parameter for nonlinearity; following change in their values it can provide fast detection of any anomaly that can be subject of different drivers/excitations/interaction of the medium.

2. "Are there any observations of solitary structures in the ionosphere? Observational evidence would be good to mention."

Answer: We have cited few observed evidences for the obtained solution:

Hallinan T. J., & Davis T. N. 1970, Planet. Space Sci. 18, 1735 for different dynamical morphologies of the aurora.

We can provide some more references regarding this topic as:

D. A. Gurnett, R. L. Huff, J. D. Menietti, J. L. Burch, J. D. Winningham and S. D. Shawhan 1984, J. Geophys. Res. 89, A10, 8971,

P. R. Fagundes, V. G. Pillat, M. J. A. Bolzan, Y. Sahai, F. Becker-Guedes, J. R. Abalde and S. L. Aranha 2005, J. Geophys. Res. 12302, A10, 8971.

Lin C. H., Wang W., Hagan M. E., Hsiao C. C., Immel T. J., Hsu M. L., Liu J. Y., Paxton L. J., Fang T. W., & Liu C. H. 2007, Geophys. Res. Lett. 34, L11112

and

Huang C. Y., Marcos F. A., Roddy P. A., Hairston M. R., Coley W. R., Roth C., Bruinsma S., Hunton D. E. 2009, Geophys. Res. Lett. 36, L00C04

Both references discuss the anomaly in density either electron or neutral; our solution obtained for the potential as bright soliton (enhanced), means that the density is decreasing within the soliton size since the particles are trapped in that potential. If the potential is dark soliton, then it can explain enhanced density within the same region.

As far as simulations are concerned, we have cited the paper that clearly provide evidence for the structures predicted by our solution:

Maruyama, N., Y.-Y. Sun, P. G. Richards, J. Middlecoff, T.-W. Fang, T. J. Fuller-Rowell, R. A. Akmaev, J.-Y. Liu, & C. Valladares, 2016, Geophys. Res. Lett. 43 (6), 2429

3. "In addition, major drivers of the ionosphere, including the solar irradiance, space weather conditions, and lower atmospheric forcing are not considered in the study at all. "

Answer: The aim of the paper is not to discuss the trigger of the ionospheric disturbances, but rather to propose the solution for the possible structure as a result of that disturbance. It is not always possible to create such structure. The aim of the paper is to re-investigate the condition for that creation. It turns out that assumption of the constant unperturbed

thickness of the layer, used in number of papers discussed the ionosphere dynamics using wave phenomena, is not adjusted.

4. "The electrodynamical effect in the ionosphere is not included either."

Answer: We have solved self-consistently fluid equations and Maxwell's equation for somewhat simplified case, using electromagnetic filed. Effects of electric filed enter the equation of motion via conductivity, due to collisions of neutrals and charged particles. We have already suggested to add appendix with derivation of ExB drift/ponderomotive force acting on the fluid.

We hope that Referee will be satisfied by answers. Please, do not hesitate to ask if there are any further questions.

Changes made in the manuscript

Regarded comments of both Referees following changes have been made in the manuscript:

1. Page 2: In order to relate our solution with the simulations of dynamics of the ionosphere we have cited Maruyama et al. 2006 in previous version but in this revised one we have underlined the obtained density peak structure. Also, we have cited the experimental work of van Hejist and Kloosterzeil 1989 to underline the importance of rotation on the solution existence.
2. Page 4: Detailed derivation of equation of motion Eq. (2) has been mentioned in Appendix A. It has been done to confirm the existence of the induced electric field resulting in the conducting fluid. In Appendix (page 14), we have cited Schunk and Nagy 2009 recommended by the Referee and related our derivation and approximation with ones given in the reference.
3. Page 10: In the first sentence, we have underlined observation of Huang et al 2009 mentioned in Section 1, as a direct consequence of enhanced scalar potential derived in our work in which that particles are trapped.
4. Page 10/11: We have started section Discussion with all relevant instabilities that could possible cause perturbations which can result in solitary structure with reference Immel et al. 2006 suggested by Referee. Also, it has been cited few papers that point out observed importance of horizontal transport that is crucial in our research: Haldoupis and Pancheva 2002, Tsunoda et al. 1998, Forbs and Leverony 1992.
5. Page 13: We have added appendix with detailed derivation of momentum equation for weakly ionized ionosphere description. Conductivity is parameter that involve induced electric filed.

All changes are marked by bold letters except Appendix A which is included as it is. Appendix B is in revised version remains the same as in previous just it has been renamed by B.